# On Approximating the *pIC*_50_ Value of COVID-19 Medicines In Silico with Artificial Neural Networks

**DOI:** 10.3390/biomedicines11020284

**Published:** 2023-01-19

**Authors:** Sandi Baressi Šegota, Ivan Lorencin, Zoran Kovač, Zlatan Car

**Affiliations:** 1Department of Automation and Electronics, Faculty of Engineering, University of Rijeka, Vukovarska 58, 51000 Rijeka, Croatia; 2Faculty of Dental Medicine, University of Rijeka, Krešimirova 40/42, 51000 Rijeka, Croatia

**Keywords:** artificial neural networks, convolutional neural networks, machine learning, *pIC*
_50_, regression modeling, SMILES

## Abstract

In the case of pandemics such as COVID-19, the rapid development of medicines addressing the symptoms is necessary to alleviate the pressure on the medical system. One of the key steps in medicine evaluation is the determination of $pIC_{50}$ factor, which is a negative logarithmic expression of the half maximal inhibitory concentration ($IC_{50}$). Determining this value can be a lengthy and complicated process. A tool allowing for a quick approximation of $pIC_{50}$ based on the molecular makeup of medicine could be valuable. In this paper, the creation of the artificial intelligence (AI)-based model is performed using a publicly available dataset of molecules and their $pIC_{50}$ values. The modeling algorithms used are artificial and convolutional neural networks (ANN and CNN). Three approaches are tested—modeling using just molecular properties (MP), encoded SMILES representation of the molecule, and the combination of both input types. Models are evaluated using the coefficient of determination ($R^{2}$) and mean absolute percentage error (MAPE) in a five-fold cross-validation scheme to assure the validity of the results. The obtained models show that the highest quality regression ($\bar{R^{2}}=0.99,\;\sigma_{\bar{R^{2}}}=0.001;\;\bar{MAPE}=0.009\%,\;\sigma_{\bar{MAPE}}=0.009$), by a large margin, is obtained when using a hybrid neural network trained with both MP and SMILES.

## 1. Introduction

When a global pandemic arises, there are many different types of pressure put on the healthcare system, mostly caused by a large number of infected patients that require treatment and quarantine to lower the mortality and spread rates of the disease at hand [1]. This was most clearly seen in the recent COVID-19 pandemic. The quick development of not only vaccines that will control and lower the spread of the disease [2], but also of the medicines which will allow for the treatment of the infected is key in the combat of pandemics [3]. Due to the nature of the pandemic and similar epidemiological diseases, treatment may need to be developed rapidly to lower the healthcare system and societal impacts of such an event [4]. Still, the development of pharmaceuticals targeted at combating a particular disease is a daunting task even without the pressure of a looming pandemic [5]. For this reason, the creation of tools that may help alleviate the issues related to the process may be key in combating arising diseases.

One of the tools commonly used for general rapid development is artificial intelligence (AI) [6], particularly the data-driven methods commonly referred to as machine learning (ML) [7]. AI-based modeling techniques have shown wide applications in many fields, including medicine and related scientific branches [8], and were shown to be particularly useful in addressing various issues faced during the recent pandemic [9]. While not necessarily the be-all and end-all of modeling, especially in sensitive environments such as patient care, AI-based tools are shown to be very good preliminary design tools [10,11], which serve to assist experts in the given field in making more precise, better informed, and most-importantly, faster decisions [12].

One of the key processes in pharmaceutical development which may be addressed by an AI-based tool is the determination of the half maximal inhibitory concentration ($IC_{50}$) [13]. The usual determination of $IC_{50}$ requires the determination of the dose–response curve through the examination of how the different concentrations of antagonist affect the reversal of agonist activity [14]. This can be a lengthy and complicated process. For this reason, the authors propose the usage of an AI-based artificial neural network (ANN) for the initial approximation of the $pIC_{50}$ value which is the negative logarithm of the $IC_{50}$ expressed in molars. The goal is to develop a system that will, based on the molecular makeup of the certain compound, approximate its $pIC_{50}$, which may allow the pharmaceutical professional to decide whether testing the compound at hand further is necessary or not. Due to how fast the ANNs are at calculating the output value, this could allow for wider testing of different compounds.

From a large amount of recent research, it can be concluded that the matter of predicting drug parameters, especially $IC_{50}$, is a widely researched and interesting topic. To assure the novelty, the presented research will focus on applying a custom new type of model (hybrid combination of CNN and MLP ANNs), with the goal of increasing the performance of the models beyond the previously achieved ones, as shown above. The researchers posit the following research questions (RQ):RQ.1. Can ANNs be used to approximate the $pIC_{50}$ values, especially on a low amount of data?RQ.2. Can the above approximation be successfully performed using just molecular properties (MP), just SMILES annotation of the compound, or are both necessary?RQ.3. If the above approximation is possible, which are the hyperparameters of the ANN model that can provide the best results for the task?

The description of the used dataset will follow, with the authors performing a basic statistical analysis to determine the dataset quality and the modeling approach. Then, a description will be given of the dataset preparation for its use in the aforementioned techniques, followed by an explanation of different ANNs used in the modeling process. Finally, the results obtained with the described methodology will be presented and discussed with the appropriate conclusions drawn from them.

## 2. Materials and Methods

In this section, the dataset is briefly described, followed by the description of different methods used to precisely regress the output.

### 2.1. Dataset Description

The research is based on a publicly available dataset entitled “COVID-19 Drug Discovery Data”, provided by the Indian Government. The dataset was released in 2020, for the purpose of analysis of drugs that were considered as possible treatments for alleviating symptoms of COVID-19, based on the medicines that were being provided to the patients at the time as well as the medicines discussed as possible treatments at a governing level [15]. The dataset was provided as a list of the compounds, with their SMILE notation, and was further expanded with molecular properties based on the PubChem library of chemical compounds [16]. The dataset is publicly available from [17], while the additional chemical details of the compound can be looked up in the aforementioned PubChem database. Observing the chemicals, it can be noticed that all of the compounds describe organic molecules of a relatively large size. The dataset in question consists of a SMILE notation of the compound, chemical descriptors/properties of the molecule (to be referred to as molecular properties-MP, in the rest of the paper), and the $pIC_{50}$ value of the individual molecule. The goal of the dataset is to provide data for the development of the models which approximate the $pIC_{50}$ of the compound based on the existing data. A total of 104 compounds that have been used to treat COVID-19 patients are contained in the dataset.

Simple data preparation is performed before the data is used in further research, for machine-learning model training. Some of the data points do not have a numerical value for $pIC_{50}$, instead replacing it with the term “BLINDED”. Due to these data points not being usable, they are removed from the dataset, yielding a dataset with 94 data points. Some of the data points are missing certain molecular properties within the data. To avoid losing additional data points in an already relatively small dataset, a sentinel value of −1 is used to fill the empty cells [18]. As no original data has the value of −1, meaning that the sentinel value directly indicates the non-existent data in the data vector. In addition to row removal, some of the columns are removed, such as compound identifying information (name or PubChem compound ID), or other non-numerical data such as alternative molecule descriptors that cannot be generally processed within the dataset. The descriptive statistics of the values that have been kept within the dataset are given in Table A1, within Section A.1. As the table shows, after the described dataset preparation process the following molecular descriptors are kept within the dataset, in addition to the SMILES molecule description and $pIC_{50}$ value of the compound: molecular weight, XLogP [19], exact mass, monoisotopic mass, topological polar surface area, complexity, charge, H-bond donor count, H-bond acceptor count, rotatable bond count, heavy atom count, isotope atom count, atom stereo count, defined atom stereo count, undefined atom stereo count, bond stereo count, defined bond stereo count, undefined bond stereo count, covalent unit count, 3D volume, 3D X-steric quadrupole value, 3D Y-steric quadrupole value, 3D Z-steric quadrupole value, 3D feature count, 3D feature acceptor count, 3D feature donor count, 3D feature anion count, 3D feature cation count, 3D feature ring count, 3D feature hydrophobe count, 3D conformer model RMSD, 3D effective rotor count, and 3D conformer count. The distributions of all the used numerical data points are given in Figure A1, within Section A.2, with the best fitting distribution to each given in the subfigure. What can be noted is that none of the data vectors follow the normal or uniform distribution. This fact, in addition to the relatively low number of data points, points to the fact that additional validation, such as k-fold cross-validation, will be necessary to properly validate the obtained data-driven models [20]. In addition to that need, the descriptive statistics show that certain inputs are constant (isotope atom count, charge, undefined bond stereo count, defined atom stereo count). Due to the inputs being constant through the entire dataset, they should have no influence on the output [21]. Observing the median and standard deviation values of the variables, it can be seen that other variables have a large value spread, which is confirmed by the data distribution histograms. Observing the correlation of the individual properties to the $pIC_{50}$ output, it can be noted that all of the correlation values of the inputs are relatively low, with none having an absolute value of the correlation higher than 0.5. Covariance is also generally low, except for certain variables such as complexity (−44.903), and exact and monoisotopic masses (−16.863 and −16.883, respectively).

In addition to the above-discussed variables, a SMILES notation [22] of each compound is provided. This allows the research to utilize the molecular makeup of the compound itself as one of the inputs in the modeling. However, to achieve this, the SMILES notation needs to be transformed in a manner that allows its use as an input for an ANN. To achieve this, a system is developed which will allow for the transformation of the SMILES string into a one-hot encoded matrix [23]. One-hot encoded matrices are given a commonly used way of transforming a string of symbols into a matrix format [24].

To transform the SMILES string into the desired format, the algorithm will go through the entire dataset and split SMILES strings into individual symbols. While doing this, each unique symbol will be noted and stored in the memory. This first iteration will serve to determine which SMILES symbols exist in the dataset. An additional element that will be noted is the maximum symbol length of the SMILES notation string (which is not necessarily equal to the number of characters, due to elements such as “Br” representing a single symbol consisting of two characters [25]). The number of unique symbols and the maximal SMILES length will allow us to determine the size of the one-hot matrix to be used. The matrix will have the number of rows equal to the number of unique symbols and the number of columns equal to the maximum SMILES length. An example of such a matrix is given in Figure 1. In the illustrated example, the dataset would consist of molecules that consist of the following symbols “C”, “-”, “Cl”, and “Na”, with the maximum SMILES symbol length of 10. When applied to a realistic set of molecules, such as the dataset that is being observed in the presented research, the individual matrix will be larger. When the first step of the conversion has been performed a total of 21 unique symbols are found in the dataset (in order of appearance: “Cl”, “C”,“1”, “=”, “(”, “N”, “O”, “)”, “S”, “2”, “F”, “#”, “3”, “[”, “+”, “]”, “-”,“4”, “Br”, “I”, “\\”), with the maximal SMILES symbol length of 78. This approach of adjusting the input matrix size has the benefit of avoiding unnecessary padding by making the matrix size large enough to fit all the possible SMILES symbols. In addition to the previous concern, using a fixed-size matrix introduces the issue of needing to determine the maximum possible size of the molecule. As the generation of SMILES one-hot encoded matrices are relatively fast [26] and a changed matrix size only requires a minor adjustment to the input layer of the neural network, the authors suggest that generating new one-hot encoded SMILES matrices for new datasets is a more practical approach.

Once the algorithm has passed through the entire dataset and determined all possible unique symbols and maximal symbol length in the dataset, the proto-matrix which was created in the previous step needs to be filled and stored for later use. The developed algorithm will once more iterate over all of the SMILES strings. The algorithm will then iterate over each symbol in the string. It will place a value of 1 in the column that is equal to the position of the symbol in the string, in that row whose value is equal to the symbol. The rest of the rows in the column will be filled with zeroes. This process will repeat for each symbol until there are no more symbols left in the SMILES string. If there are columns left in the matrix they are filled with zeros [27]. An example of the filled matrix is given in Figure 2. The reason for this padding is that ANNs which are to be used in the later modeling steps require all of the inputs to have uniform dimensions [28].

The prepared matrix is then appended with the parameters and the output value in the general shape of (dim(parameters),(rows×columns×depth),output), with the total dataset being given in the tensor of the shape (N,(dim(parameters),(rows×columns×depth),output)), where *N* is the number of elements [29]. For our case, the dimension of the whole dataset is as follows: (94,(33,(21×78×1),1)). The dataset in question is then stored in the appropriate NPY format for further use in the regression modeling process. The full algorithm, as just described, is also provided in pseudocode using the set notation in Algorithm 1.
**Algorithm 1** The algorithm for transformation of SMILES strings into SMILES matrices**Require:** D
▹ SMILES dataset    U←Ø▹ Unique symbols    m←0▹ Maximal symbol length    s←""▹ Empty symbol    **for each** SMILES∈D **do**
          **for each** s∈SMILES **do**
                **if** s∉U **then**
                       $U^{⌢}s$
                **end if**
                **if** dim(SMILES)>m **then**
                       m←dim(SMILES)
                **end if**
          **end for**
    **end for**
    $T_{dim(U)\times m}$▹ One-hot matrix to be filled    T←Ø▹ Transformed dataset    P←Ø▹ MP values from dataset    y←0▹ Output value    **for each** SMILES∈D **do**
          i←0▹ For tracking vector columns          **for each** s∈SMILES **do**
                $P\leftarrow D_{P}$
                $y\leftarrow D_{y}$
                **if** s∈U **then**
                     $\mathrm{Let}\;x\;\mathrm{be}\;\mathrm{such}\;\mathrm{that}\;U_{x}=s$
                     $T_{x,i}=1$
                     $\forall x^{\prime }\in dim\left(U\right)\wedge x^{\prime }\neq x\Rightarrow T_{x^{\prime },i}=0$
                **else**
                     **for each** xin[1,dim(U)] **do**
                           $T_{x,i}=0$
                     **end for**
                **end if**
                i←i+1
                T←(P,T,y)
          **end for**
    **end for**


### 2.2. Neural Network Regression

There are three approaches tested with neural networks, depending on the data that is used in each separate case. Each of the three neural network architectures used corresponds to one of the three data configurations. The architectures used are:Only SMILES encoded as a one-hot matrix;Only molecule parameters;The combination of both SMILES encoded as a one-hot matrix and molecule parameters.

The individual architectures will be discussed in further subsections. All of the architectures are custom to the problem at hand due to the input matrix size. For each architecture, three different hyperparameters are adjusted—the number of epochs the model is trained for [30], the batch size of the data used for the training [31], and the solver algorithm used for the model training [32]. The hyperparameter values are adjusted using the grid search (GS) method, which means that each possible combination of the three hyperparameters is tested. The tested hyperparameter values are given in Table 1.

The data is split into training and testing sets in an 80:20 ratio. The ratio is selected due to the application of the five-fold cross-validation process, as mentioned in the previous section [33]. Each of the ANNs is trained in the same manner with the forward and backward propagation process. In this process, each individual data point is brought to the input of the ANN. Then, the matrix containing the input data *I* is multiplied with the individual weight matrices [34], or convoluted with the weight filters in the case of the convolutional neural network (CNN) [35]. The values in these matrices/filters are initially set to random values. This will lead to a predicted output at the end of the ANN. If we denote this output with $\hat{y}$, then the entire vector of all predictions for each of the inputs in the dataset can be expressed with $\hat{Y}$. The corresponding vector of real values, in this case, the $pIC_{50}$ values of the dataset, can then be expressed as $Y=[y_{1},y_{2},\cdots ,y_{n}]$. This allows us to calculate the vector of errors ℰ according to the root mean square error (RMSE) formula which was used for the model training in this research [36]:(1)$$E\leftarrow \varepsilon_{n}={\left(y_{i}-\hat{y_{i}}\right)}^{2}\forall y_{i}\in Y_{i},\hat{y_{i}}\in \hat{Y_{i}}$$

The mean value of the error represents the general error of the dataset in the given epoch. The loss function is then calculated as the mean of the dataset errors $J\left(W\right)=\bar{E}=\frac{1}{N}\sum_{i=0}^{N}\varepsilon_{i}$, where *N* is the number of the data points in the training set (75 or 76 depending on the dataset fold used for evaluation), and W is the set of the weights [37]. This loss is then backpropagated through the dataset where the values of the weights W are then adjusted depending on the value of the loss function, as [38]:(2)$$W_{e+1}=W_{e}-\frac{\alpha }{N}\cdot \frac{\partial J(W)}{\partial W},$$
where *e* is the current training epoch, and α is the learning rate, which is set as the default value for each of the used solvers [39].

The code in this paper is implemented in Python 3.9.12 programming language. The tensor manipulation that was previously described for transforming the dataset with one-hot encoding was performed using the NumPy library version 1.23.0. The ANNs were designed and trained in Tensorflow version 2.9.1. The score evaluation is performed using the Scikit-Learn library, metrics submodule, version 1.1.1.

#### 2.2.1. ANN for Regression Based on Molecule Properties

The first ANN used in the research is shown in Figure 3. It utilizes only the 33 MPs as an input and is constructed as a standard multilayer perceptron (MLP) [40]. This means that the network consists of the first layer (which has a size equal to the number of inputs-33), one or more hidden layers, and an output layer consisting of a single neuron whose value is equal to the value of the ANN output [38]. The base architecture (layer configuration) of this and the following networks have been based on the conclusions of existing research in the field [23,41,42,43,44,45]. The used network has a total of four hidden layers consisting of 32 neurons for the first and second hidden layers, 64 neurons total for the third hidden layer, and 32 neurons in the last hidden layer. All of the layers are densely connected, meaning that each neuron in a given layer has a weighted connection to all neurons in the subsequent layer. All of the layers use the rectified linear unit (ReLU) [46,47] activation function, given as F(x)=max(0,x).

#### 2.2.2. CNN for Regression Based on SMILES

The second type of the neural network applied is a CNN that is meant to create a model that regresses the $pIC_{50}$ model based solely on the SMILES one-hot encoded matrix. The reasoning is that the ANN should be capable of determining the information derived from the SMILES in the shape of MPs themselves. Such an approach would save the time needed to generate and storage space needed to store the extra info for the MP. The CNN was shown to be a high-performing algorithm on tensor-shaped data [48].

The input to the network is the same as the previously stated image size (21×78×1). Then, a series of three stacks of layers is repeated. Each stack consists of a two-dimensional convolutional layer, batch normalization, activation, and a two-dimensional maximum pooling layer. The two-dimensional convolutional layer performs the convolution operation between the input of the layer (for the first layer this is the input matrix) and a filter tensor [49]. The sizes of the filters in the used CNN are 3×3×64, 2×2×128, and 2×2×256, respectively. The values within the filter tensors are originally set to random values and then adjusted according to the previously described training process [50]. The second element of each of the three stacks is batch normalization. This layer normalizes its input for each of the training batches. This allows for faster training due to the easier reparametrization of the model [51]. The next layer in each of the stacks is the activation layer, which applies the activation function ReLU to the entirety of the previous layer output [52]. Finally, max pooling is applied. This technique takes the value of tensor elements in the m×n grid and converts them to a single element which is equal to the maximal value of the elements [53]. This lowers the computational cost of the training and introduces a basic invariance to the internal (detailed) representation of the data in the model [54]. The sizes of the max pooling filters used in the research are 3×3, 2×2, and 2×2, in order of the application. After the three stacks, the flatten layer is applied, which takes the final tensor (shaped 1×6×256) and transforms it into a vector [55]. This vector is then used in the same manner as the input used in the MLP described in the previous section. This layer is then densely connected to a single hidden layer of 33 neurons, which is in turn connected to the layer with a single neuron that will serve as the output of the CNN. The CNN model is fully shown in Figure 4.

#### 2.2.3. CNN for Regression Based on SMILES and Molecule Properties

The final variant of the ANN the research applies to the data is a hybrid CNN model combining both of the previously described ANNs, the MLP and CNN. This combination will allow for the processing of both the SMILES matrices and numerical MP data. The idea behind this application is to combine both the input types and in doing that provide more information to the learning framework of the ANN, in the hopes of increasing the regression quality.

As shown in Figure 5, the first part of the network, up to the last three hidden layers are equal to the ones described in the two previous subsections. The last layer of the two architectures does not end in a single neuron but instead ends with a densely connected layer of 32 neurons. Then, a concatenate layer [56] is utilized to stack both of the 32-element vectors into a single 64-element vector. The combined outputs of two neural networks are then densely connected to a pre-final layer of 32 neurons. This layer is finally densely connected to the final output layer of a single neuron, the value of which will represent the output of the hybrid ANN, as was the case with the two previously described ANNs.

### 2.3. Result Evaluation

The results are evaluated using two metrics—the coefficient of determination ($R^{2}$) and mean absolute percentage error (MAPE). $R^{2}$ is a metric that has a range from <0,1.0> and shows the amount of variance explained between the real output set $Y=[y_{1},y_{2},\cdots ,y_{n}]$ and the predicted output set $\hat{Y}=\left[\hat{y_{1}},\hat{y_{2}},\cdots ,\hat{y_{n}}\right]$, where *n* is the number of the elements in the output vectors [57]. The higher the $R^{2}$ values are, the more of the variance is explained, meaning a higher quality regression model [58]. $R^{2}$ is calculated according to [59]:(3)$$R^{2}=1-\frac{\sum_{i=0}^{n}{\left(y_{i}-\hat{y_{i}}\right)}^{2}}{\sum_{i=0}^{n}{\left(y_{i}-\bar{y}\right)}^{2}}.$$

The other metric used is MAPE, which is the mean of absolute differences between corresponding elements of the output sets *Y* and $\hat{Y}$, expressed as a percentage [60]. Due to it being expressed as a percentage, it can easily be used to compare the precision of different models. This metric is calculated as [61]:(4)$$MAPE=\frac{\sum_{i=0}^{n}\frac{y_{i}-\hat{y_{i}}}{y_{i}}}{n}$$

Because the dataset is relatively small, five-fold cross-validation has been applied. As seen in Figure 6, the original dataset is split into the dataset uniformly randomly without repetition, with each of the subsets containing an equal number of data points [62]. With the five subsets, the process of training can be started. In each of the five repetitions, a different subset is used as the testing set, while the remaining four subsets are mixed to create the training set [63]. Each of these recombined training-testing datasets are referred to as a fold. A prediction score is calculated on each of the training folds. These five scores are then used to calculate the average of the scores across each of the folds, with the standard error. The goal of this procedure is to avoid models that will overfit on individual data points, as a well-generalizing model will have good scores on each of the subsets [64].

For the purposes of this research, the model is considered to have satisfying performance if it fulfills the following two criteria, which have been determined depending on the review of the previous research in the field (Table 2):Condition 1. achieves the $R^{2}$ score higher than 0.99 when accounting for the lower bound of scores according to the standard error $\sigma_{R^{2}}$ across all five testing folds;Condition 2. achieves the MAPE error lower than 1.0% when accounting for the higher bound of errors according to the standard error $\sigma_{MAPE}$ across all five testing folds.

## 3. Results and Discussion

The results in Figure 7 show the performance of the best-performing model according to the $R^{2}$ scores, for each of the tested input variations. The best model using MP has achieved the $R^{2}$ score of 0.54±0.36, while the model using SMILES achieved a slightly better and significantly more stable score of 0.66±0.04. According to the previously set condition 1, neither of the models fulfilled the score required for them to be considered satisfactory. The model combining both input types achieved a significantly higher $R^{2}$ score of 0.99±0.001. This score is not only very high but is also very stable across folds indicating a robust model. We can conclude that this model satisfies the previously set condition 1.

When MAPE scoring of the best-performing models for each input type is observed as presented in Figure 8, we can note that only the model which combined SMILES and MP as input satisfied condition 2. The model based on the MP inputs has achieved the MAPE of 1.83%±0.95. Interestingly, the best-performing model when only SMILES are used as an input shows a poorer performance than when evaluated using $R^{2}$, which supports the need for multi-metric evaluation as performed. The SMILES-based model has achieved a MAPE of 3.98%±0.11, which still shows it as a more stable model than MP, although with poorer MAPE performance. Finally, the model based on both SMILES and MP inputs achieved a comparatively low error of 0.009%±0.009, again indicating it as the only model to satisfy condition 2.

Table 3 shows the hyperparameters of the best-performing models for each of the input variations. Due to the same models having the best performance when evaluated with both metrics in all cases, only a single set of the models is given with scores corresponding to the ones in Figure 7 and Figure 8. All the models used different solvers, with relatively low batch sizes. Interestingly, the number of epochs is relatively low for all the models, with the best-performing model achieving the best results in only five epochs. The low number of epochs needed to achieve the result in question, combined with the low standard error of the best-performing model indicate that overfitting did not present a significant issue.

Due to all of the input variations having used the same hyperparameters during the GS procedure, another comparison can be performed. The average scores across all the tested hyperparameters can be compared to determine the overall performance of the tested architectures as described in the previous section. This allows us to determine whether the architecture designed for a given set of inputs had a comparatively good performance overall. This is important for further application of the developed architectures in future research, as it allows us to observe the robustness of the architectures and determine whether the architecture achieved good performance on only a few hyperparameters or did it perform well overall. This comparison is given in Table 4.

Table 4 shows that the overall best-performing models have been ones that used both SMILES and MP as inputs. The model that used only MP as input is shown as the worst in comparison to the other two. An interesting thing to note can be observed when the scores for the models using only SMILES are viewed. While the overall performance of SMILES-only models is poorer than the models which combined the two used inputs, the models obtained using only SMILES show significantly smaller average percentile errors across folds. This indicates that these models have a tendency to be more stable on different data, which is an important thing to note for future research.

### Result Comparison to State-of-the-Art Research

Masarweh and Darsey (2022) [75] showed the application of functional correlation and NETS AI-based software to determine the modified $IC_{50}$ values. The prediction is based on the molecule energy values and focuses on targeting the creation of Alzheimer’s disease medicine. Cho et al. (2022) [65] demonstrate the extraction of numerical features from cell images exposed to tested drugs. The goal of the research is to determine the $IC_{50}$ value without the need for the staining process. The authors use a linear relationship coefficient ($r^{2}$) and achieve a precision of between 0.94 and 0.95. Zheng et al. (2022) [66] focuses on the determination of a different factor, Cluster of Differentiation 93 (CD93), as a predictor of the molecular subtype and therapy response, with a focus on bladder cancer. The authors perform a classification task and achieve the area under the receiver operating characteristic curve (AUC) of 0.808. Begum and Parvathi (2022) [67] use quantitative structure–activity relationship (QSAR) models to predict cathepsin L (CTSL) inhibition due to SARS-COV-2 virus causing a decrease in the factor. Authors achieve an $R^{2}$ score of 0.663. Lee and Nam (2021) [68] utilize CNN architectures AlexNet and GoogLeNet for predicting the $IC_{50}$ value for cell growth inhibition in cancer patients. The results are evaluated using AUC in micro- and macro-configurations due to multiple classes. The achieved results range between 0.58 and 0.98, depending on the configuration and the targeted dataset. Shishir et al. (2022) [69] demonstrate the prediction of new drug properties, specifically $IC_{50}$ using graph CNNs. The authors apply cross-validation on the dataset and achieve an $R^{2}$ score of 0.52. Rajput et al. (2021) [70] demonstrate an application of various methods in order to predict $IC_{50}$/$EC_{50}$ of medicines repurposed for COVID-19, from ’DrugRepV’ database. The authors apply the support vector machines (SVM), k-nearest neighbors (KNN), and ANNs. The achieved results are evaluated using Pearson’s correlation coefficient and range between 0.60 and 0.90. Jin and Nam (2021) [71] propose HiDRA-Hierarchical Network for Drug Response Prediction with Attention, which is an interpretable AI-based model which can be used for predicting drug responses in cancer cells, molecular pathways, and drug levels. HiDRA system shows a prediction RMSE of 1.0064, Pearson’s correlation coefficient of 0.9307, and an $R^{2}$ value of 0.8647. Immidisetty and Agrawal (2021) [72] apply pre-constructed AI-based solutions chemically interpretable graph interaction network version 2 (CIGIN2) and deep learning model for solvation-free energies in generic organic solvents (DELFOS) on the problem of prediction the solvation free energy. They achieve a MAE of 0.5323, RMSE of 1.304 and MAPE of 28.21%. Gong et al. (2020) [73] have demonstrated the application of 2D- and 3D-QSAR methods, SVMs, extra tree (ET) regressor, random forest (RF), ridge regressor, KNN, and others. The authors apply the aforementioned methods to the investigation of drugs addressing diabetes mellitus, and achieve the best results of 0.98 when evaluated using $R^{2}$, with ET regressor. Hermansyah et al. (2021) [74] use deep learning (DL), gradient-boosted trees (XGBoost), RF, MLR, and SVMs to construct QSAR models achieving an $R^{2}$ score of 0.922. An overview of the discussed articles is given in Table 2.

Comparing the best-achieved results from the presented research, $R^{2}=0.99\pm 0.001$ and MAPE=0.009%±0.009, for the model combining SMILES and MP data, we can see that the results are satisfactory. Compared to the best results regarding $R^{2}$, namely [73] which achieved $R^{2}=0.98$, it can be noted that the result is slightly better than what the authors achieved. It should be noted that authors in [73] did not use cross-validation, which could result in poorer results. Compared to the other research which used $R^{2}$ for the evaluation [67,69,71,74] the presented models achieved significantly better results showing $R^{2}$ improvement in the range between 0.068 and 0.47. Even the poorer scoring models, based on MP and SMILES input data, achieve higher scores than some previously published research, such as [69], the only of the compared papers that used MAPE, allowing for the direct comparison is [72]. As seen in the Table 2, the authors achieved the MAPE in the range of 28.21 at the lowest, and 36.63 at the highest. This shows a significant improvement, compared to the best results achieved by the SMILES + MP model. Even the poorer results, achieved by MP and SMILES-based models by themselves show improvement in comparison to the results achieved by the authors.

## 4. Conclusions

The goal of the presented research was to determine the possibility of using the AI/ML-based ANN algorithms to derive models for the approximation of the $pIC_{50}$ factor of various medications contained in a publicly available dataset.

(RQ.1.) The achieved results show that ANNs can be used to approximate the value in question, in some configurations, with the best-performing model achieving scores that satisfy both conditions set (lower bound of $R^{2}>0.95$, higher bound of MAPE<1.0%). The achieved scores indicate that the model in question has a high performance, and the cross-validation results performed indicate that the model is robust on the entirety of the data available for validation. (RQ.2.) The achieved scores show that the regression model of satisfactory quality can only be achieved when a custom architecture that combines the MP and SMILES inputs is applied. Still, observing the performance across the hyperparameters tested in GS shows that the model based on only SMILES has the ability to achieve models which are extremely stable across the various hyperparameter ranges, although performing poorly in comparison to the models based on MP and SMILES inputs. This characteristic is something that should be considered in further research. The achieved results indicate that such research should focus on methods combining molecular notation and properties, with the possible addition of models which are based on just molecular notation (SMILES). Based on the research results, models based on only the MP of compounds may not provide satisfactory performance, at least not without significant changes to the ones presented here. (RQ.3.) The hyperparameters of the models show that a large number of epochs are not necessary to achieve good results, with the longest training to achieve the best results being 25 epochs out of the maximum 300, for the MP-based model. The same can be concluded in the case of batch sizes. The maximum, in this case, is the batch size of 8 in the case of the models based on SMILES + MP data. As the maximum batch size tested was 64, this indicates that such a wide range of hyperparameters as used in the presented research is not necessary. No regularity can be seen in the solver selection. If the future research hyperparameter range (e.g., for training the architectures on a larger dataset) was limited to the next largest hyperparameter tested after the largest hyperparameter that was used to achieve the best-performing models, a significant decrease in training time would be achieved. With the described training setup only 240 models would be trained, compared to the 672 in this research. Especially considering that the number of epochs is one of the most influencing hyperparameters for model training, time-savings could be significant. Some possible concerns exist with the presented research. The first of which is the low number of data points in the dataset. This concern arises from the wish to focus the research on a relatively small dataset, to examine if the presented methods could be used on a lower number of compounds that are being developed in order to address a particular disease (such as the mentioned COVID-19). In the suggested case, a low number of pharmaceuticals may be available for further training. The concern was addressed via the application of a cross-validation technique to test the performance of the models further and ascertain their robustness as much as possible. Another concern is the lack of architecture variation beyond the tested hyperparameters. The authors designed the layers and activation functions of the networks based on past research and experience. While it is possible that better results could have been achieved with larger networks, this specific issue was not the main concern of the research. As mentioned, the results show that the main research goal of developing a model for precise approximation of the $pIC_{50}$ factor on the provided dataset has been achieved fully.

As presented, the paper has some limitations which need to be acknowledged. The main limitation of the paper is the limited size of the dataset used for the modeling, at only 94 data points. While this has been partially addressed with the application of cross-validation, it has to be noted that this may influence the prediction quality on the larger dataset. Another thing to note is that the developed models have not been externally validated, meaning that their performance has not been tested outside of the limited dataset used in the research. These steps are something that would have to be addressed before the application of the system as developed in real medicine development. Finally, it has to be noted that the dataset used for research contains relatively old data, and significant research effort has been expanded since widening the knowledge regarding COVID-19 and medications. So, while the presented paper can be used as a proof of concept, indicating the possibility of applying the described techniques to determining the $pIC_{50}$ of COVID-19 medicines, the research should be further expanded using not only a larger number of but also more modern medicines in the future.

Future work may focus on testing the developed models on different datasets consisting of MP and SMILES, with the goal of regressing the $pIC_{50}$ value, in order to determine which further tuning may be necessary. In this case, special focus should be given to the process of transfer learning and cross-testing between different sets of compounds. This process would allow us to determine whether the models can be applied to different data while keeping a satisfactory performance and if transfer learning can be applied for further model tuning in case the performance needs to be improved. 

## Figures and Tables

**Figure 1 biomedicines-11-00284-f001:**
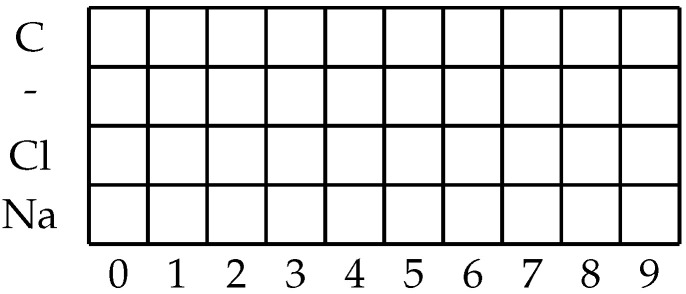
An example of the matrix prepared for the one-hot encoding for the dataset with the maximal SMILES symbol length of 10 and 5 unique symbols.

**Figure 2 biomedicines-11-00284-f002:**
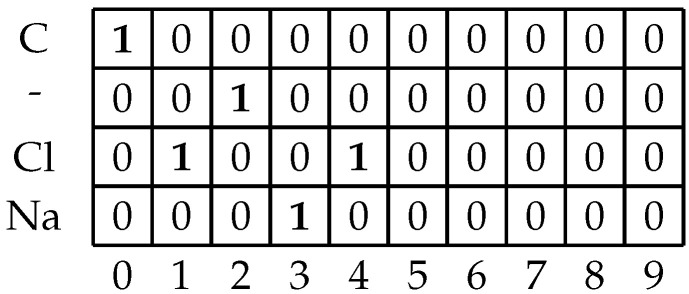
An example of the prepared one-hot matrix containing an example SMILES molecule “CCl-NaCl”. Ones bolded for emphasis.

**Figure 3 biomedicines-11-00284-f003:**
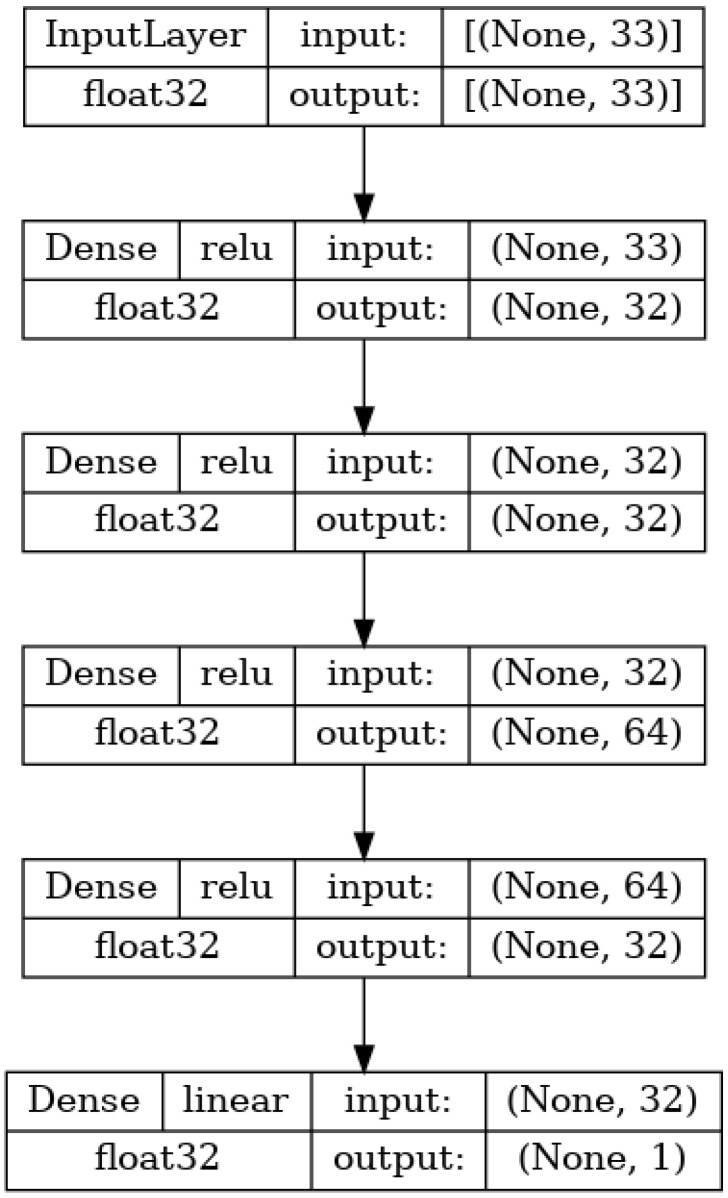
Architecture of the molecule property-based regression ANN.

**Figure 4 biomedicines-11-00284-f004:**
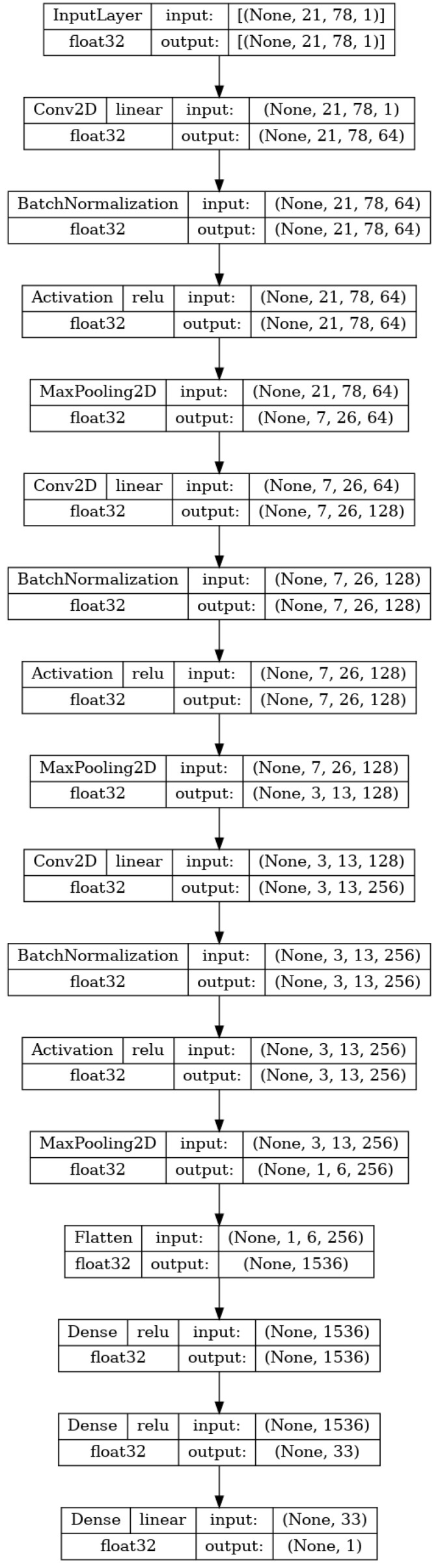
Architecture of the SMILES-based regression CNN.

**Figure 5 biomedicines-11-00284-f005:**
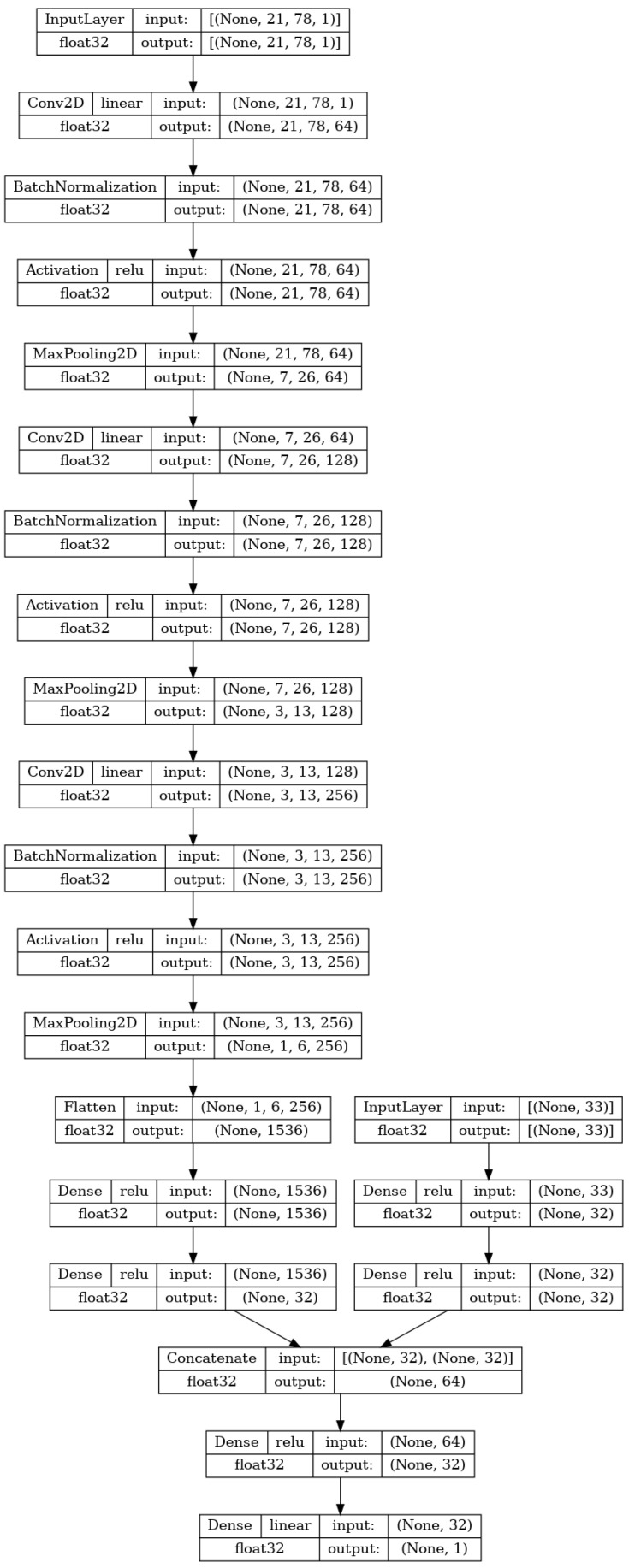
The regression CNN architecture combines the one-hot encoded SMILES and molecule properties.

**Figure 6 biomedicines-11-00284-f006:**
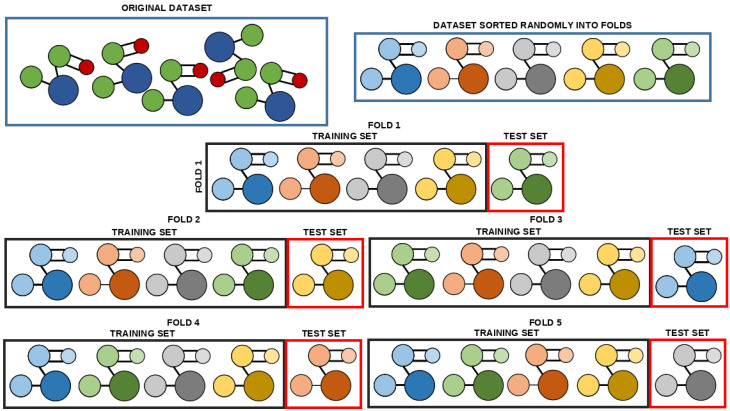
The illustration of the five-fold cross-validation procedure used in the research.

**Figure 7 biomedicines-11-00284-f007:**
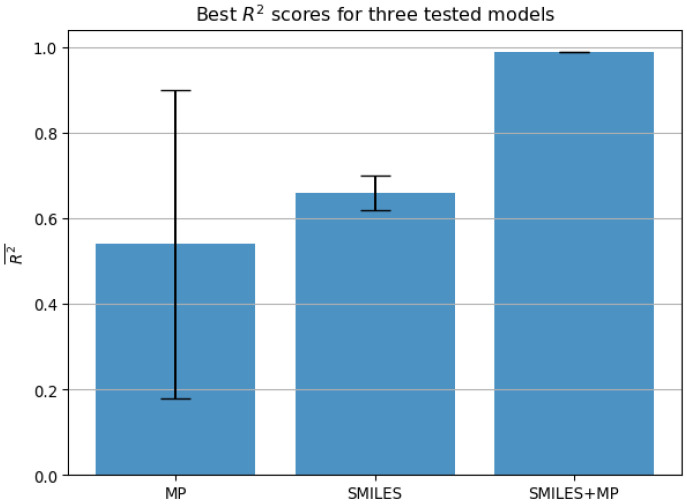
The best results achieved with each of the neural network configurations, evaluated using $R^{2}$ metric (higher is better), expressed as mean across folds and standard error (MP—molecular properties, $R^{2}$—coefficient of determination).

**Figure 8 biomedicines-11-00284-f008:**
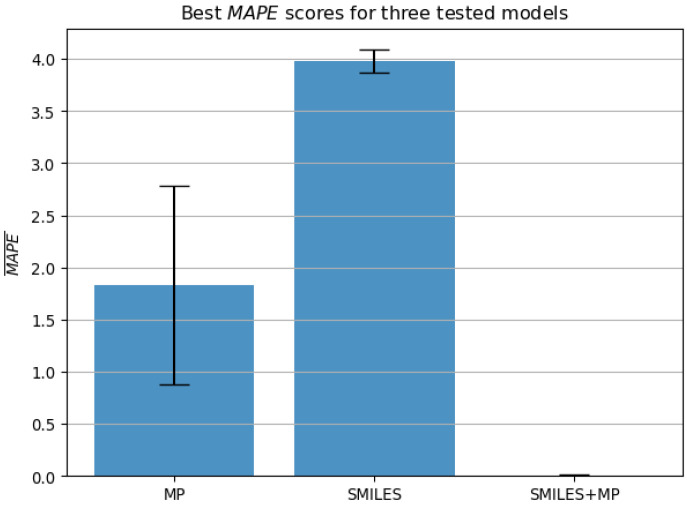
The best results achieved with each of the neural network configurations, evaluated using MAPE metric (lower is better), expressed as mean across folds and standard error (MP—molecular properties, MAPE—mean absolute percentage error).

**Table 1 biomedicines-11-00284-t001:** Hyperparameter values used in the GS.

Hyperparameter	Possible Values	Count
Batch size	1, 2, 4, 8, 16, 32, 64	7
Epochs	1, 2, 5, 10, 25, 50, 75, 100, 150, 200, 250, 300	12
Solver	SGD, RMSProp, Adam, Adadelta, Adagrad, Adamax, Nadam, FTRL	8
Total	672	

**Table 2 biomedicines-11-00284-t002:** Overview of the state-of-the-art research published in the field in the last two years.

Paper Reference	Prediction Goal	Method	Most Notable Results Achieved
[65]	$IC_{50}$	FE	$r^{2}\in <0.94,0.95>$
[66]	CD93	ANN	AUC=0.808
[67]	CTSL	QSAR	$R^{2}=0.663$
[68]	$IC_{50}$	AlexNet, GoogLeNet	AUC∈<0.58,0.98>
[69]	$IC_{50}$	Graph CNN	$R^{2}=0.52$
[70]	$IC_{50}$/$EC_{50}$	SVM, KNN, ANN	r∈<0.60,0.90>
[71]	$IC_{50}$	HiDRA	RMSE=1.0064
			r=0.9307
			$R^{2}=0.8647$
[72]	Solvation free energy	CIGIN2, DELFOS	MAE∈<0.5323,0.5615>
			RMSE∈<1.178,1.304>
			MAPE∈<28.21,36.64>
[73]	$IC_{50}$	SVM, ET, RF, Ridge, KNN	$R^{2}=0.98$
[74]	$pIC_{50}$	DL, XGBoost, RF, MLR, SVM	$R^{2}=0.922$

**Table 3 biomedicines-11-00284-t003:** Hyperparameters of the best-performing models for each of the observed input cases.

	Batch Size	Epochs	Solver
MP	1	25	Adadelta
SMILES	2	10	FTRL
SMILES + MP	8	5	Adam

**Table 4 biomedicines-11-00284-t004:** The average results for N = 672 tested architectures in GS. (MP—molecular properties, $R^{2}$—coefficient of determination, MAPE—mean absolute percentage error, GS—grid search).

	${\bar{R^{2}}}^{\mathrm{GS}}$	${\bar{\sigma_{R^{2}}}}^{\mathrm{GS}}$	${\bar{\mathrm{MAPE}}}^{\mathrm{GS}}$	${\bar{\sigma_{\mathrm{MAPE}}}}^{\mathrm{GS}}$
MP	0.135173025	0.748035001	4.269612071%	2.450460031%
SMILES	0.633779051	0.027208207	3.990937597%	0.097102445%
SMILES + MP	0.723649767	0.55142646	0.120139103%	0.201233967%

## Data Availability

Publicly available datasets were analyzed in this study. This data can be found here: https://www.kaggle.com/datasets/divyansh22/drug-discovery-data/metadata, accessed on 16 January 2023.

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
