# Peer review of "On Approximating the pIC50 Value of COVID-19 Medicines In Silico with Artificial Neural Networks"

_biomedicines, 2023, doi:10.3390/biomedicines11020284_

Round 1
Reviewer 1 Report
The manuscript entitled On approximating the pIC50 value of COVID-19 medicines in-silico with artificial neural networks" is an interesting and current topic. I am sure this type of analysis will help in new drug discovery. However, this manuscript has some potential flaws
1. The details of the molecules missing. From where these molecules came and their credentials including experimental data?
2. Which techniques the authors have adopted for cross-validation?
3. Also, the database the author used is two years old when we had little knowledge about COVID-19.
Author Response
We would like to thank the Reviewer for their review of our manuscript. Please find the answers to the questions posed below. The changes made in the manuscript due to this reviewer's comments have been marked with red.
The details of the molecules missing. From where these molecules came and their credentials including experimental data?
The following text was added to further help with explaining the provenance of the used dataset:
“The dataset was released in 2020, for the purpose of analysis of drugs that were considered as possible treatments for alleviating symptoms of COVID-19, based on the medicines which were being provided to the patients at the time as well as the medicines discussed as possible treatments at a governing level [15]. The dataset was provided as a list of the compounds, with their SMILE notation, and was further expanded with molecular properties based on the PubChem library of chemical compounds [16]. The dataset is publicly available from [17], while the additional chemical details of the compound can be looked up in the aforementioned PubChem database. Observing the chemicals, it can be noticed that all of the compounds describe organic molecules of a relatively large size, as is to be expected.”
Which techniques the authors have adopted for cross-validation?
The authors have applied a five-fold cross validation, which was clarified with the addition of the following text, along with the newly added figure 6:
“Because the dataset is relatively small, five-fold cross-validation has been applied. As seen in figure 6, the original dataset is split into the dataset uniformly randomly without repetition, with each of the subsets containing an equal number of data points [59]. With the five subsets, the process of training can be started. In each of the five repetitions, a different subset is used as the testing set, while the remaining four subsets are mixed to create the training set [60]. Each of these recombined training-testing datasets are referred to as a fold. A prediction score is calculated on each of the training folds. These five scores are then used to calculate the average of the scores across each of the folds, with the standard error.”
Also, the database the author used is two years old when we had little knowledge about COVID-19.
This comment was noted in the limitations section of the paper - a newly added segment of Conclusion section, as follows:
“Finally, it has to be noted that the dataset used for research contains relatively old data, and significant research effort has been expanded since, widening the knowledge regarding COVID-19 and medications. So, while the presented paper can be used as a proof of concept, indicating the possibility of applying the described techniques to determining the $pIC_{50}$ of COVID-19 medicines, the research should be further expanded using not only a larger number of, but also more modern medicines in the future.”
Kindest regards,
Authors
Reviewer 2 Report
In this study the authors use ANN and CNN to approximate pIC50 values, relying only on the molecular makeup of drugs (using the SMILE notation and/or molecular properties).
It is an interesting study showing how AI methods can be used to predict several pharmacological properties (in this case, pIC50) necessary for optimizing pharmacotherapy.
Comments:
1. My main concern is the limited dataset used for model building. Only 94 cases were available, and this may affect the predictive ability of the model. The authors are advised to discuss this issue as a limitation of their study.
Also, external validation is needed to verify the model's predictability.
2. The Introduction section can be shortened. There is too much information and presentations of other studies. I suggest moving lines 51-88 to the Discussion section and discussing them in comparison to your results.
3. Please mention the software and the packages (e.g., if it was done in Python) you used for the development of ANNs and CNNs
Minor comments:
1. Table 2, describing the main statistical properties of the dataset, is of secondary importance and can be moved to the Appendix.
2. Figure 1 can also be moved to the Appendix. Also, fonts cannot be read now and need to be enlarged.
3. Figure 6 needs to be larger since it is very difficult to read.
Author Response
We would like to thank the Reviewer for reviewing our manuscript. Please find the answers to each of the individual questions posed below. The changes made in the manuscript due to this reviewer's comments have been marked with light blue.
My main concern is the limited dataset used for model building. Only 94 cases were available, and this may affect the predictive ability of the model. The authors are advised to discuss this issue as a limitation of their study. Also, external validation is needed to verify the model's predictability.
The following text was added to the Conclusion section in order to address the limitations of the research:
“As presented, the paper has some limitations which need to be acknowledged. The main limitation of the paper is the limited size of the dataset used for the modeling, at only 94 data points. While this has been partially addressed with the application of cross-validation, it has to be noted that this may influence the prediction quality on the larger dataset. Another thing to note is that the developed models have not been externally validated, meaning that their performance has not been tested outside of the limited dataset used in the research. These steps are something that would have to be addressed before the application of the system as developed in real medicine development.”
The Introduction section can be shortened. There is too much information and presentations of other studies. I suggest moving lines 51-88 to the Discussion section and discussing them in comparison to your results.
The authors have moved the text which was previously at the lines 51-88, and the accompanying table (previously table 1, currently table 4). We have added a new subsection 3.1. “Result comparison to state-of-the-art research” at the end of section 3 “Results and discussion”. In addition to moving the text and the table, we have added the following text in order to compare the achieved results to the previously published ones:
“Comparing the best-achieved results from the presented research - R2 = 0.99 ± 0.001 and MAPE = 0.009% ± 0.009, for the model combining SMILES and MP data, we can see that the results are satisfactory. Compared to the best results regarding R2, namely the [71] which achieved R2 = 0.98, it can be noted that the result is slightly better than what the authors achieved. It should be noted that authors in [71] did not use cross-validation, which could result in poorer results. Compared to the other research which used R2 for the evaluation ([65], [67], [69], and [72]) the presented models achieved significantly better results - showing R2 improvement in the range between 0.068 to 0.47. Even the poorer scoring models, based on MP and SMILES input data achieve higher scores than some previously published research, such as [67]. The only of the compared papers that used MAPE, allowing for the direct comparison is [70]. As seen in the table 4, the authors achieved the MAPE in the range of 28.21 at the lowest, and 36.63 at the highest. This shows a significant improvement, compared to the best results achieved by the SMILES+MP model. Even the poorer results, achieved by MP and SMILES-based models by themselves show improvement in comparison to the results achieved by the authors.”
(Please note, to indicate the change more clearly, we have marked the title of the newly added subsection, and the newly added text blue, while we have left the text that was only moved from the introduction (previously lines 51-88) black.)
Please mention the software and the packages (e.g., if it was done in Python) you used for the development of ANNs and CNNs
The following text was added to add the requested information:
“The code in this paper is implemented in Python 3.9.12 programming language. The tensor manipulation that was previously described for transforming the dataset with one-hot encoding was performed using the NumPy library version 1.23.0. The ANNs were designed and trained in Tensorflow version 2.9.1. The score evaluation is performed using the Scikit-Learn library, metrics submodule, version 1.1.1.”
Table 2, describing the main statistical properties of the dataset, is of secondary importance and can be moved to the Appendix.
Table 2 has been moved to the Appendix A.1.
Figure 1 can also be moved to the Appendix. Also, fonts cannot be read now and need to be enlarged.
The Figure 1 has been moved to the Appendix A.2., and its size has been increased to fill the entire page, hopefully making it more readable.
Figure 6 needs to be larger since it is very difficult to read.
The size of Figure 6 has been increased. In order to adjust the formatting of the paper and avoid white space, the size of other figures depicting the neural network architectures has also been increased and they were slightly moved.
Kindest regards,
Authors
Round 2
Reviewer 1 Report
The authors updated the manuscript with the clarification. I believe the current form of the manuscript is acceptable for publication.